# Information provision to caregivers of children with rare dermatological disorders: an international multimethod qualitative study

Carleen Walsh [1], Gerard Leavey [2], Marian McLaughlin [3]

¹Bamford Centre for Mental Health and Wellbeing, Ulster University, Derry, UK
²Director of Bamford Centre for Mental Health and Wellbeing, Ulster University, Derry, UK
³School of Psychology, Ulster University, Derry, UK

**Correspondence to**
Dr Carleen Walsh;
walsh-c33@ulster.ac.uk

## ABSTRACT

**Objective** To identify service-provided information needs among dermatological caregivers of patients living with ichthyosis.

**Design** This is the first online international qualitative study to explore caregiver-reported service-provided information needs, using transnational focus groups (n=6), individual interviews (n=7) and in-depth emails (n=5). NVivo facilitated the coding process and Framework Analysis was applied.

**Setting** Caregivers were recruited through two online ichthyosis support groups and resided across 10 countries and 5 continents (USA, Greece, Netherlands, Ireland, UK, Canada, India, Philippines, Switzerland and Australia).

**Participants** A purposive sample of 8 male and 31 female caregivers participated (mean age range 35–44 years). Participants were aged 18 years old or older and fluent in English. Participants cared for a total of 46 children (1:1 ratio for child gender and clinical classification of disease severity). Participants represented all stages along the care continuum, including neonatal intensive care unit and bereavement.

**Results** This study advances understanding of how to optimise information-sharing across hospital, community and online settings at three points along the care continuum (screening, active caregiving and survivorship). Timely, personalised and appropriate service-provided information support was considered key in influencing the self-efficacy, coping ability and psychosocial well-being of both the caregiver and their child. The modification of information support, through feedback loops, can result in a different bidirectional psychosocial impact for the caregiver and the affected child.

**Conclusion** Our findings provide a novel insight into how existing gaps between caregiver expectations and needs, in terms of information support, can be addressed. As information support is a modifiable factor, improved healthcare education around these themes should become an urgent public health matter to inform future educational and psychosocial interventions.

## STRENGTHS AND LIMITATIONS OF THIS STUDY

⇒ This is the first international multimethods qualitative study to explore service-provided information support from the perspective of rare dermatological caregivers.

⇒ Online recruitment from the community may have allowed the representative cohort of caregivers from across five continents to give more candid accounts of their access to information from formal services.

⇒ Distribution of the semi-structured interview schedule to consenting participants in advance of data collection promoted active participation and enhanced depth of inquiry.

⇒ The purposive sampling strategy may negatively influence the transferability of findings to non-users of online peer support groups.

⇒ The under-representation of male caregivers and exclusion of caregivers who were not fluent in English during data collection are limitations of this study.

## INTRODUCTION

Ichthyoses is a group of over 20 rare, chronic, inflammatory skin diseases characterised by thickened, scaling and dry skin.[1] It is a highly visible, life-limiting, incurable, often fatal disease, for which there is limited healthcare knowledge and few treatment options available to caregivers.[2] Ichthyoses can exert a substantial burden of care, potentially impacting the quality of life for both those affected and their caregivers.[3] Dermatology Life Quality Index scores place ichthyoses among the skin disorders with the most harmful impact on a patient's quality of life.[4] Although the nature of ichthyosis means that global management is symptomatic, episodic, unpredictable and often time-consuming,[4] caregivers are often obliged to assume nursing roles without prior training or appropriate information. Although the recently proposed Supportive Care Needs Framework[5] for parents caring for a child with a rare disease includes information support, no qualitative study or dermatology Core Outcome Set has explored the information needs of any group of rare dermatological caregivers to date.[6 7] This is important given

that an information tension may exist around the provision of medical care when discordant approaches exist to providing information on the management of symptoms, emphasising the role of vigilant protector and increasing the risk of burnout.[8]

This paper will explore the information needs of caregivers of people with ichthyosis and how to optimise information-sharing across hospital and community settings at three points on the ichthyosis care continuum: (1) screening, (2) active caregiving and (3) survivorship. This qualitative study was conducted as part of a larger international project aimed at identifying a set of core outcomes for ichthyosis towards the development of the first needs assessment e-tool for informal caregivers of children living with ichthyosis.[9]

## PARTICIPANTS AND METHODS
### Study design
An online international qualitative study to explore dermatological caregiver-reported service-provided information needs, using transnational focus groups, individual interviews and in-depth emails. The conduct, design and reporting of this study follows the Standards for Reporting Qualitative Research (online supplemental file 1).[10]

### Patient and public involvement
An expert group was established at the outset of this study and were actively involved in the design (research notice, participant information sheet, consent form and interview schedule), methodology and reporting of this research. Consultant dermatologists, based in Europe and the USA, initially suggested several leading international academics, healthcare professionals, policy advisors and support group administrators as potential participants for the expert group. This group included consultant dermatologists who both lead and run trials, but who were also practicing clinicians. To ensure that the methodology was informed using caregiver experts, two caregivers involved in academia were included in the expert group.

Caregivers of children with ichthyosis were the main information resource for this qualitative study. Caregiver checking and peer debriefing was conducted to assess the quality of research findings, whereby participants were invited to comment on our interpretation of feedback via return email.

### Study population and recruitment
Participating caregivers were aged 18 years old or older, fluent in English and provided daily care for a child (no age restrictions) diagnosed with any ichthyosis subtype during the previous 15 years. Care was defined as any care over and beyond what is considered normal for a typically developing child.[11] Participants were recruited through two medically recognised ichthyosis support groups, the ISG (Ichthyosis Support Group),[12] and the FIRST (Foundation for Ichthyosis and Related Skin Types),[13] who

posted the research notice online. This notice contained an embedded Participant Information Sheet link, which contained an embedded consent link, created using Qualtrics. Due to disease rarity, the estimated sample size for this study was 30–35 caregivers. The guiding interview schedule was emailed 1 week in advance of initial contact (online supplemental file 2) and participants could indicate their preferred method of data collection. Informal emails were sent to all potential members of the expert group describing the aim and objectives of the project and requested a return email specifying whether they were willing to provide emailed consent to support the study.

### Data collection
To establish trustworthiness,[14] the semi-structured interview schedule was developed on an ad-hoc basis by a group of multidisciplinary professionals (n=13), who held extensive knowledge and proven clinical experience of ichthyosis. Test interviews were undertaken with volunteer caregivers (n=13) to assess the clarity of content and wording, relevance, number and grouping of questions.[15] High inter-coder agreement between the caregiver and expert group confirmed that the questions were consistently understood or interpreted in relation to the research question.

Due to the pandemic and participants' geographical spread, online audio focus groups (n=6), individual interviews (n=7) and in-depth emails (n=5) were used. In all, 43 caregivers were recruited and 39 were interviewed. Interviews stopped once data saturation had been reached and no new themes emerged. Focus groups contained between four to seven participants, with a mean duration of 85 min. Interviews had a mean duration of 64 min. Data collection was audio-recorded and transcribed verbatim (CW). Flexibility in the interview schedule enabled new issues to be followed and/or clarify feedback (online supplemental file 2). The interviewer (CW) had training and experience with in-depth interviewing.

### Analysis of qualitative data
Framework Analysis was deemed the most suitable approach to address our research question as it emphasises how both a priori issues and emergent data driven themes should inform the development of the analytical coding framework.[16] The thematic framework was mainly based on a priori determined issues. To ensure the coding framework would be open to novel or unanticipated data, the first two focus groups were double coded (CW and GL) and agreement was high between coders. Any discrepancies were resolved through consensus with all authors. NVivo (V.10) facilitated the coding process. Most themes were expected but 'online medical support groups' was identified as an emergent theme from the inductive encoding process. For increased transparency, the final analytical coding framework arising from the NVivo analysis is clearly presented in online supplemental file 3.

## RESULTS

### Characteristics of the sample

Thirty-one female and eight male informal caregivers (mean age range 35–44 years) participated. Caregivers included one grandparent, three adoptive parents, one neonatal intensive care unit parent, one bereaved parent, two affected (patient) caregivers and three parents of adult children (18–25 years). Six parents had more than one affected child. All other caregivers were parents of an affected baby, preschool/school-aged or teenager. Participants resided across 10 countries and 5 continents (USA, Greece, Netherlands, Ireland, UK, Canada, India, Philippines, Switzerland and Australia).

Caregivers provided care for 46 children, with a 1:1 ratio for both child gender and clinical classification of ichthyosis severity (autosomal recessive congenital ichthyosis subtypes (n=17) vs less severe subtypes (n=29)). Eight different subtypes of ichthyosis were represented in this study: X-linked, ichthyosis vulgaris, lamellar ichthyosis, epidermolytic ichthyosis, Harlequin ichthyosis, Netherton's syndrome, congenital ichthyosiform erythroderma and ichthyosis en confetti.

A detailed overview of caregivers and care recipients is presented in online supplemental file 4.

### Themes

The five key themes (online supplemental file 5) that capture caregivers' experiences of service-provided information are outlined below at three points along the care continuum: (1) screening, (2) active caregiving and (3) survivorship. The themes and subthemes relating to optimal service-provided information, was shown to be primarily influenced by the point on the care continuum and to a lesser extent by information source (online supplemental file 5). Indicative quotations from caregivers (Q1–77), relevant to each of the identified points along the care continuum, are included in tables 1–5.

### Screening

#### Theme 1: genetic diagnosis and counselling

Caregivers expressed a need for prenatal testing and/or automatic referral for timely diagnosis of ichthyosis. Those who received prenatal testing perceived an advantage in both future family planning and accessing of timely specialist medical care (Q1 and Q2). Timely postnatal genetic diagnosis reduced caregiver uncertainty and facilitated referral for the best available treatment, important for improving long-term medical, psychological and social outcomes. Two caregivers reported experiencing profound ambivalence and conflict about disclosure. This conflict oscillated depending on their simultaneous desire to maintain hope, primarily achieved through denial, temporality and the construction of different potential disease outcomes and their desire to share relevant information with the wider community (Q3 and Q4).

Although several caregivers experienced biographical disruption at the time of diagnosis, most caregivers preferred transparent information on prognosis and treatment options. Genetic counselling was significant during the early stages, helping to reduce feelings of shame, denial, grief and anxiety. This helped them to accept, modify, mould and adjust caregiving roles and relationships more quickly (Q5 and Q6). In turn, this positively affected their ability to achieve personal goals

| Table 1 | Indicative quotations from screening stage | | | |
|---|---|---|---|---|
| **Point on care continuum: screening** | | | | |
| Quote number | Caregiver ID code | Sex | Country | Quote |
| 1 | 2 | F | UK | The temptation to google was so great, so actually having some solid information from them was really beneficial and sort of grounded us. |
| 2 | 35 | F | USA | Yeah information right, knowledge, that's a big one. I had never heard of ichthyosis before. If you have that diagnosis prenatal or postnatal, it's nice to have an information sheet as well as contact information. I think it should be automatic, like a dermatology referral is put in immediately before you leave the hospital just so that you could be armed with that knowledge so that you know the best way to care for your baby. |
| 3 | 19 | F | NI | But I think for me anyway the main thing would have been getting, well with my oldest, being diagnosed straight away because it was clear that he had ichthyosis from when he was born but I knew nothing about ichthyosis. |
| 4 | 38 | F | Greece | The only thing I think got me through the first bit when we didn't know what her diagnosis was that we didn't know how severe it was going to be or had no clue what was going to go on. |
| 5 | 29 | F | Switzerland | This tiny little thing in front of you has no idea how to tell you what it needs, so yeah definitely initially having information was invaluable. |
| 6 | 12 | F | USA | We did have the genetic testing done with him when he was like one or two and that was really helpful to then find out. |
| F, female; M, male; NI, Northern Ireland. | | | | |

**Table 2** Indicative quotations from active caregiving (hospital level)

| Point on care continuum: active caregiving<br>Service provided information at hospital level | | | | |
| --- | --- | --- | --- | --- |
| Quote number | Caregiver ID code | Sex | Country | Quote |
| 7 | 29 | F | Switzerland | The nurses have taught me that I should ask for help, that it is my right to ask for it and that they are trying to teach me about that it is not all about (child), it is about the family as a whole, the ability to give her care. Because if I am not in a good space, physically, mentally, then I cannot care for her. |
| 8 | 28 | F | Canada | He hasn't seen a dermatologist since I'd say maybe in about 6 years, and you have to go through a doctor to get a referral for a dermatologist here. The doctor won't refer because obviously you're doing what you're supposed to be doing and they're not going to tell you any different so, but I think it would be good to have a record. |
| 9 | 38 | F | Greece | I've asked for referrals to a skin specialist, to a dermatologist and you know she just said, oh no he looks fine, you don't need that but you know now at this point, we're seeing as he grows that his ichthyosis is more apparent. |
| 10 | 23 | F | Australia | When (child) went to (named) hospital, there was a dermatologist and wound care nurse specialist when she was in neonatology. The paediatric community nurses would contact them for advice and the nurses at varying times would go to wound care conferences and meet up with experts. |
| 11 | 31 | M | Ireland | And the relief to see her in a hospital bed with people who had an understanding of what was going on. Her skin was repairing. They were actively doing, understanding, researching, trying to get around her. It gave me a huge relief. |
| 12 | 13 | F | USA | We were very lucky that she was born in (hospital) and they got help from an expert in the University. It was a great support to us because she knew from early on what we were dealing with, failure to thrive and her skin was peeling every day and obviously in pain, so that was good to have at the beginning. |
| 13 | 5 | F | UK | I think for us, medical professionals that understand or have some sort of knowledge, or if they don't have the knowledge to say that they don't have the knowledge and that they'll go and find somebody who does. |
| 14 | 3 | F | Ireland | The ichthyosis nurse was here with us teaching us how to do the bandaging and the creams and stuff like that and they supported us once a week for quite a while until we became confident in her care. |
| 15 | 39 | M | Ireland | The only change I would have liked was when (child) died. It was all very quick and soon after she passed, we were being asked to get ready to leave. We had to ask for information. We asked would they wrap her in a blanket for the trip. I just couldn't drive with her in the car and felt they were very surprised that I was leaving her there until the next morning. She died about 10pm. |
| 16 | 31 | M | Ireland | I used to get into the hospital, and I had a very strict nurse who was actually wonderful, but she was very strict and rightly so. In at nine, bathe him, do all his creams, do his bandages. |
| 17 | 16 | M | USA | It is just that there is such little information out there and community nurses and parents just want a protocol of care. |
| 18 | 23 | F | Australia | It's being that carer and having access to information which can help you be the best. You've got to be constantly thinking about their care needs. |
| 19 | 26 | F | Ireland | In more recent years, it was actually since he had his episode with wanting to kill himself, he got referred to a pain specialist in (hospital), who has since left his post because the resources were not there and there is now no pain specialist in the (country). |

F, female; M, male.

around establishing effective care structures and planning for the future. The perceived validation of disease symptoms among healthcare staff and receipt of appropriate disease-specific information facilitated coping and access to healthcare services. In contrast, the relative lack of access to genetic diagnosis and counselling in developing countries negatively impacted access to medical expertise and future family planning.

**Table 3** Indicative quotations from active caregiving (community level)

| | | | | |
|---|---|---|---|---|
| **Point on care continuum: active caregiving**<br>**Service provided information at community healthcare level** | | | | |
| Quote number | Caregiver ID code | Sex | Country | Quote |
| 20 | 1 | F | UK | Our health visitor was also really good and she was researching ichthyosis herself and she was in touch with the ISG and the hospital and getting information so that she could support us. She put us in touch with the local community nursing team because I wasn't coping very well. |
| 21 | 4 | F | Ireland | I remember making a call at work and I couldn't make the call. It was just a normal phone call. My heart was racing and then I literally had to hang up. I did it a few more times and I thought I can't function. I went to the GP. She was amazing, and said we've all been waiting for you to come in here. I've had to go through counselling. It's hard work. |
| 22 | 27 | F | Ireland | By the time you leave the NICU a few days later, you forget about everything that was at your disposal, you know. You didn't need it in those early days. You needed it once you were settled at home. I guess that was on me to follow it up, who was now home, exhausted and without any time. Like I said, I didn't even know where to call. At one point I tried to reach out to somebody and had a wrong number. |
| 23 | 29 | F | Switzerland | Once they seen that she was very young, so there was still much wound care dressing involved, they actually asked what they could do to help us. I am very lucky. Here, they recognise this condition and they do provide support. They call the service 'Y' which basically means hospital care in the home. What would I do without them? I think I would have lost my mind. |
| 24 | 29 | F | Switzerland | Just the physical nature of it is really something, and these challenges make me feel very tired quite often. And I think what tends to happen is whenever I am particularly tired or particularly stressed that anger rises up and it is hard to control that feeling. |
| 25 | 36 | F | Netherlands | I think sometimes when you just get stuck in your situation, you don't realise there can be anything else out there, and you kind of forget to ask for yourself. Someone should tell you this is what's available or this is what you can try. |
| 26 | 2 | F | UK | I went to the doctor at one stage. He told me I needed to exercise and eat better. That was his answer. I didn't have any bloody time to exercise or eat better. |
| 27 | 3 | F | Ireland | You nearly need someone to ring you up and say that you need to come to this thing. It's as important as you making sure you get sleep at night and it's as important as you making sure you've washed your clothes or that you have put underwear on. You know what I mean. It actually is that important. |
| 28 | 26 | F | Ireland | I made the time and mustered up the nerve to go and see the GP to ask for help, and twice I was offered antidepressants. I just never took that up. I just don't see how that is going to help in the long run and I have quite an addictive personality so I can't afford to go down that road. |
| 29 | 27 | F | Ireland | She couldn't understand me anymore, she couldn't understand the situation, she was visiting us within a role where she had to work inside the box and have a list of procedures to activate if things were happening with the child or if a parent wasn't responding to the guidelines she was giving,. So our relationship got extremely fraught because I had to fight and say that she was missing the point here. Our child isn't going to fit onto those charts. It's not going to fit into that box. It's not going to respond. So it was a huge moment in my head when I decided that I would have to stop taking the advice from (hospital) and find my own way and try and dip into different things. |
| 30 | 25 | F | Philippines | The main caregiver is always the mother and the family. There is no real outside support. |
| 31 | 32 | F | India | So I decided to bring her to a dermatologist here in my country, but as you know we are in a third world country. Not everybody here knows about ichthyosis. The child was terribly malnourished, and the doctors said just breastfeed the child some more. I did for hours every day. What the doctors were unable to recognise was the fact that my child was unable to suck on the breast due to the contracted skin. |

**Table 3** Continued

| Point on care continuum: active caregiving | | | | |
|---|---|---|---|---|
| **Service provided information at community healthcare level** | | | | |
| 32 | 32 | F | India | We came to hear of a little boy in a village close by in India in Bengaluru who had ichthyosis. They wanted to chase the mother and the child out of the village because they said that the mother had bad genes. We decided to go to the village. I took (child) there and when we went there, the villagers were so shocked to see her as a grown young woman who was able to talk in their language and also that she was a perfectly normal girl, but with a skin condition. She spoke with them and the villagers decided they were going to keep the child and the mother. |
| 33 | 32 | F | India | Now ichthyosis is not commonly seen here in India, and I emphasise on the word seen. It does not mean that it is uncommon, but it does mean that it is not seen. Consanguine marriages are quite common here and result in many cases of ichthyosis, but these cases are either hidden away or they disappear. Very, very few cases are seen. |
| 34 | 1 | F | UK | Successful inclusion was down to the sharing of information between the medical professionals, families and the schools. |
| 35 | 12 | F | USA | They sent him home for the entire week until he got a doctor's note stating that he could come back, because they were convinced that it was contagious. |
| 36 | 26 | F | Ireland | They just had not got the empathy or compassion to understand to slow down or the change the game or to not leave him out in the yard. So I took him out of school as it was coming to a kind of a crisis point. We now home-school. |
| 37 | 21 | F | USA | As soon as they heard that they would have to apply cream they said that (child) needed to go to a special needs school. |
| 38 | 26 | F | Ireland | He has high anxiety and has sleep issues that are massive. He has mental health issues from the stigma he experienced at school. A lot of different things feed into it. His itch, his pain, the blistering, the cracking, the visibility of his skin and the psychological impact of that and feeling isolated. |

F, female; GP, general practitioner; ISG, Ichthyosis Support Group; M, male; NICU, neonatal intensive care unit.

## Active caregiving

### Theme 2: service-provided information at hospital level

Availability of service-provided information support at hospital level differed between developed and developing countries and was dependent on formal recognition of ichthyosis by national healthcare systems (Q7 and Q8). Caregivers emphasised the importance of timely referrals to dermatology expertise for appropriate information on available treatments and therapies (these were also noted to have limited data on their effectiveness) (Q9). Several caregivers described coping as a balancing act between being hopeful and well-informed (Q10). Information barriers included the rarity of ichthyosis, incompatible health beliefs, lack of clinician knowledge and/or clinical time and stigma at familial and societal level (Q11). Caregivers felt that hospital healthcare teams had a greater understanding of the physical and psychosocial impact of ichthyosis compared with community healthcare teams, suggesting the need for improved hospital-community education and training opportunities. Dermatologists, social workers, dieticians, physiotherapists, psychologists, pain specialists, occupational therapists, speech and language therapists, pharmacists, nurses and paediatricians were noted as particularly helpful in communicating appropriate information in-person and/or via online means (Q12).

When dermatological expertise was unavailable, caregivers valued hospital clinicians who were willing to source online dermatological expertise before initiating treatment to reduce potential risks to patients (Q13). Caregivers who were supported by clinicians to ask questions and request information gained confidence in their ability and felt empowered to provide appropriate care for their child from the outset (Q14). Several caregivers wanted to know whose responsibility it was to initiate discussions on accessing dermatological and genetic expertise. Regardless of disease severity, caregivers preferred when dermatologists employed a personalised or patient-centred information approach to manage subtype-specific symptoms and treatments. One bereaved caregiver emphasised the need for improved healthcare education and training on how to communicate and support families who have just experienced the death of a child in hospital (Q15). Having an assertive hospital clinician who took the lead, either in person or virtually during times of crises proved particularly beneficial (Q16). New caregivers preferred printed information relating to disease background, symptom management, treatment and practical supports. Medical care plans, co-developed and reviewed with caregivers, promoted shared decision-making and self-efficacy and were crucial for reflecting changes along the care continuum (Q17).

**Table 4** Indicative quotations from active caregiving (online support groups)

**Point on care continuum: active caregiving**
**Service provided information at online support group level**

| Quote number | Caregiver ID code | Sex | Country | Quote |
|---|---|---|---|---|
| 39 | 28 | F | Canada | I remember being horrified when he was two or 3 weeks old and sheets of skin would just peel off of him. I didn't even know that was a thing and then I learnt from other moms on this Facebook group that shedding is very much a thing. They have helped me find solutions to help with daily routines. |
| 40 | 31 | M | Ireland | …their children are a little bit older than mine and they are helping me. I want to give time back to people who are just landing into the group, or who are going through the journey I was in. |
| 41 | 4 | F | Ireland | You do take joy in every little achievement. And as I said the standards are probably lower, but other parents mightn't have as much joy in those little achievements. |
| 42 | 17 | M | USA | We're not in the same river, but we are in similar boats. You know, the feeling of not being alone in dealing with something is a huge thing. |
| 43 | 24 | F | Australia | I'm thankful for groups like this because you guys get it and I don't feel so lonely anymore. So long as I have support groups that I can find like this along the way, you know that's what helps you get through it. |
| 44 | 18 | F | Ireland | I recognised quickly that there was a lot to learn and if I opened my mind to it that that would be a good thing. That resistance at first was a hurdle for me. |
| 45 | 12 | F | USA | The mums were the ones who actually told me how to do better cares and how to deal with the problem things. |
| 46 | 25 | F | Philippines | If I can look forward to maybe a child who is maybe two or 3 years older than (child) and see them doing well, it gives me hope that (child) is going to be able to do all these things and it makes me feel better so. |
| 47 | 17 | M | USA | I will say there are, if I could call it an advantage, to having a child with ichthyosis is travelling at all. When you have a wheelchair, you get to cut the lines, pre-boarding. Look there are so many issues you have to deal with, if you get some sort of little perk, take it, right? |
| 48 | 15 | M | USA | So, you know we learnt some different ways to do things from other parents, not standing in her way, not being afraid to allow her to try it. It took a while to recognise that that was something that we had to do. |
| 49 | 26 | F | Ireland | You don't feel like you're dealing with all these battles on your own as you can feel quite lost with other friends as they kind of don't get it, the trauma as well. There's a lot of trauma to start with and other parents or people who have ichthyosis understand it form your viewpoint. |
| 50 | 11 | F | USA | Having information means we can give her the strength and the determination and realisation that she is just like everybody else. That she has the strength if she does come face to face with those negative comments, that she is strong enough to fight for herself if we're not there for her. |
| 51 | 11 | F | USA | It gives me purpose, being an advocate. The world can be really not so great. Helping people to understand what it means to live with a disability, what it means to live with a less visible or invisible disability, that is an ongoing educational necessity. You learn some new things. You learn you need to stand up for yourself. You have to stand up for your kid. And then you marvel at what they are able to do. |
| 52 | 14 | F | USA | Whereas now, I am like, hey let me tell you what my kid has. I am going to educate you because your children deserve to live in a world where children are not afraid of a different child. So personally, that advocacy is something that you learn a lot more, because ichthyosis is not a hidden disorder, it is just there. It is right there, everyone can see it, everywhere you go |
| 53 | 3 | F | Ireland | I was just trying to come to terms with the situation and it was other people's impressions and questions I couldn't deal with. I was scared because if someone asked what happened to him or what's wrong with him. I'd just burst into tears as I couldn't tell them, so I was trying to protect myself. So I didn't go out until we got answers. |

**Table 4** Continued

| Point on care continuum: active caregiving |
| :--- |
| **Service provided information at online support group level** |

| 54 | 3 | F | Ireland | When I was bringing him from hospital appointments, I was almost trying to hide him which is a terrible thing and it makes me sick to even think that, but I just couldn't bear it. |
| :-- | :-- | :-- | :-- | :-- |
| 55 | 8 | F | UK | I don't suppose me and (child) would be who we are today if we didn't have to plan around ichthyosis, so I suppose the good thing is that we are now resilient. |
| 56 | 17 | M | USA | We started a dad's Facebook group but that hasn't really taken off. As dads we still have, too many of us still have that tough façade on, I don't want to show that I am emotional. I can handle this. Everything is fine. |
| 57 | 15 | M | USA | That connectivity piece is certainly very important for me. I don't get to see these guys very often. But they are my life-long friends and they're like brothers at this stage of life. My emotional support group has been the dad's group at the conferences every 2 years, that has been the place where I could go and feel the most comfortable where you could ask any questions. |

F, female; M, male.

Continuity of care and communication by the same hospital clinician positively impacted the caregiver–clinician relationship, which appeared to promote shared decision-making, treatment adherence and caregiver self-worth (Q18). However, several caregivers reported that lack of government funding was significantly impacting on the provision of healthcare expertise (Q19).

### Theme 3: service provided information at community level

Information support at community level was primarily influenced by formal State recognition of ichthyosis, and the availability and sharing of clinician expertise between hospital and community settings. Caregivers demonstrated greater role acceptance and a greater sense of proficiency when responsive community healthcare teams proactively established links with relevant hospital teams and/or online groups pre-discharge, which served to promote community healthcare awareness and understanding of the physical and psychological needs of the family in the community (Q20). This is important given that most caregivers failed to fully realise the caregiving implications of ichthyosis until post-discharge, a time when there are fewer opportunities to access acute medical support (Q21 and Q22). Timely and appropriate service-provided information support at community level enhanced the coping skills of caregivers in the management of symptoms on discharge (Q23). Information around wound-care, itch, pain and temperature regulation in-home was noted as particularly important, with these symptoms associated with sleep disruption, negative emotions and burnout (Q24).

Community healthcare teams held privileged positions of being the caregiver's first point of contact, and generally influenced the long-term physical and psychosocial well-being of both the caregiver and child (Q25). However, most caregivers reported negative experiences when they had tried to reach out to community healthcare professionals for help (Q26). They emphasised the need for proactive community healthcare personnel to sensitively refer caregivers at regular intervals to counselling (Q27), attributing the lack of referrals to step-up expertise to limited clinical time and poor community healthcare understanding of the psychosocial impact of ichthyosis caregiving (Q28). When community healthcare teams were educated on the hidden challenges associated with ichthyosis, including delayed developmental milestones and the negative bidirectional psychosocial impact of ichthyosis between the caregiver and patient (Q29), their improved understanding of the disease promoted positive family-healthcare relationships and better enabled caregivers to prepare for the future. When dermatological support could not be accessed at community level, several acceptable trade-offs were reported. These included online peer support groups, teledermatology and designated ichthyosis liaison nurses.

Caregivers in the developing countries were more dependent on informal support, and expressed an urgent need to target service-provided information support at family and community level to reduce discrimination and social exclusion (Q30). Poor access to medical, psychological and social information contributed towards a sense of stigma, shame, anxiety, fear and loneliness for these families (Q31,Q32), which sometimes could result in the hiding, adoption or death of the child (Q33).

When effective communication pathways were established between community healthcare teams and education settings on enrolment, caregivers felt that their children experienced a more inclusive and positive mainstream education experience (Q34). Timely and appropriate communication of disease-specific and care-specific information from early intervention teams to the whole-school community reduced stigma, drop-out rates, home-schooling, discrimination and/or exclusion of the affected child (Q35–37). Consequently, this promoted caregiver return to employment, child independence

**Table 5** Indicative quotations from survivorship stage

**Point on care continuum: survivorship**

| Quote number | Caregiver ID code | Sex | Country | Quote |
|---|---|---|---|---|
| 58 | 29 | F | Switzerland | It's that kind of appliance awareness that needs to be there or that you need help with. It's what's key to setting the family up for success. |
| 59 | 29 | F | Switzerland | It was good, because it was helpful to him, and 2 hours of week of somebody coming to your house to do that was 2 hours a week that we did not have to do it. It was like a little built in break and it was great. |
| 60 | 4 | F | Ireland | I think sometimes when you just get stuck in your situation, you don't realise there can be any respite, you don't realise there can by anything else out there, and you kind of forget to ask for yourself. Someone should tell you what's available so that you can try. It allowed us to live as normal a life as possible, still work and enjoy life outside of the house, to have time to rejuvenate and have a break or whatever you need to do. For me, that meant time to visit my GP which detected my cancer in time. |
| 61 | 17 | M | USA | Also help your child to learn new skills, have somebody there who is not emotionally attached that can help them to drive them on a little bit, gently into a little bit more independence, given whatever age they're at. |
| 62 | 16 | M | USA | We have had excellent nurses that have helped us through some through some, yeah, really tough times. And that has allowed our family to stay strong, it allows our marriage to stay strong, it has been super beneficial. |
| 63 | 36 | M | Netherlands | The mental health aspect goes without saying because we're not healthy. We're doing our best but I'm sure we could be better, and we could do it without damaging ourselves quite so much. I don't feel the practical support offered was appropriate. |
| 64 | 19 | F | NI | We had to hide the knives at a point as well, where he was actually cutting himself, but the emotional support I got was for him. Was all for him. |
| 65 | 13 | F | USA | I insist on taking care of them as I am afraid that the possibility to receive inappropriate comments by others increases when their skin is not taken care of properly. |
| 66 | 17 | M | USA | That mental health support area is something that I think is really something, but we don't pay enough attention to. We certainly didn't. You're just always so busy but it's my main regret. You know I think about it and I probably should have been talking to somebody for a long time. |
| 67 | 39 | M | Ireland | I tried suicide and came very close a few months after (child) died. All I wanted was to be with (child). I pictured myself asleep lying with her. I still hate the thought of her in the grave alone. At the time I didn't have any counselling which I maybe should have had. I just didn't know what or how to deal with it. I felt that I needed to do the man thing, stay strong and work to keep things going. I was tired and worn out. Yet I was the one who had to go to work when I just wanted to stay at home. |
| 68 | 5 | F | UK | In the UK we have the NHS. I always worry about what would happen if that's not there and how you would pay. |
| 69 | 11 | F | USA | It made me less social, I go out less often and in some cases I may select more carefully the people around me and the people that come in contact with my children. |
| 70 | 26 | F | Ireland | When I'm doing her cares, it can hurt and I often go into my own bubble. I have to get it done, cause in my mind it needs to be done because it will make her more comfortable. Then when I come out of that bubble and she's like upset and she's saying I just want your skin, I just want your skin, that's when it gets to me. |
| 71 | 33 | F | Ireland | I made mistakes as I didn't have the right information. I was inflicting pain on the child all the time but that became part of what I did as her carer, so it was fine in my head. It did kind of get normalised. |
| 72 | 3 | F | Ireland | She's my wing woman that helps me out you know, and allows like myself and (partner) to have a relationship. If not I'd just be up, as in (child)'s awake every single hour so there'd be no sleep, you know. |

Continued

| Table 5 | Continued | | | | |
|---------|-----------|---|---|---|---|
| **Point on care continuum: survivorship** | | | | | |
| 73 | 29 | F | Switzerland | The nurses come and (child) can be at home where she is cooler, and no trauma happens to her skin. Then I can still give some time and attention to my other kids which is so important too. | |
| 74 | 2 | F | UK | I realised okay I'm going to go and run the toddler group and educate these people. (Child)'s going to join with her peers. Then she went from that toddler group, with her small core group of peers, onto a creche where she had the same children with her. | |
| 75 | 7 | F | UK | She comes up to the school where my older daughter is, where (child) will attend. She is in the school garden where I volunteer and the children see her, they have an awareness of her. I have done presentations in the creche where she was attending, to sharing information on how parents can approach me to talk about these issues and how they can talk to their children about it and how their children can speak to (child) about it. | |
| 76 | 12 | F | USA | His school let me go in and talk to his class, so I sat down on the floor with everybody in his class and I talked. That was the best thing that we could have done, and it made me feel a lot better. The kids knew he was different, that he needed special care. They all felt like it was their responsibility to help care for him and so minded him. | |
| 77 | 29 | F | Switzerland | I do feel quite anxious a lot of the time. I feel like it is my responsibility for her skin to be healthy all of the time and I feel like it is my fault if it's not and yes, sometimes the places that my mind goes to can…. It's not dark but I can feel quite angry sometimes. And I can sometimes feel resentful and that is not directed at her. It is not a constructive feeling. It is not going to help me. So that is my priority. To find ways to make myself feel better about it, to feel less angry sometimes. And from talking to others about this, I know there are strategies that I could use that would help make me feel less angry, or just to deal with it, perhaps just talk about it more. | |

F, female; GP, general practitioner; M, male; NHS, National Health Service; NI, Northern Ireland.

and positive inter-family relationships. Ineffective sharing and delivery of information fostered negative patient and caregiver emotions including anxiety, loneliness and anger (Q38).

**Theme 4: online information support and peer support**
When ichthyosis severity was greatest, most long-term caregivers preferred sourcing information from online peer support groups (Q39). Online subtype-specific groups were perceived as particularly helpful in accessing targeted disease and care information, while reducing the occurrence of parental downward social comparison (Q40 and Q41). Additionally, online support groups enhanced caregiver coping by reducing their emotional stress load. This was achieved by promoting a sense of self-efficacy, belonging and solidarity (Q42 and Q43) through the sharing of information relevant to symptom/behaviour management (Q44 and Q45). Caregivers highly valued the advice of online peer caregivers who were more advanced in the disease trajectory (Q46). Peer advice on reframing negative thinking and reducing negative behaviours (Q47), reduced caregiver hypervigilance (Q48) and burnout (Q49). Consequently, disease acceptance and caregiver confidence in their ability to empower their child increased (Q50). Disease acceptance encouraged caregiver advocacy, which many caregivers felt promoted confidence, self-worth, self-esteem and a sense of mastery (Q51 and Q52). Those who perceived a lack of confidence in their ability to explain ichthyosis to others living in their community socially withdrew and often hid their child to protect themselves, the child and siblings (Q53 and Q54). Caregivers awaiting access to step-up mental health expertise emphasised the invaluable role of online information in building resilience (Q55). However, a male-only focus group reported less opportunities for online information exchange due to a perceived emphasis on emotional support (Q56). Men preferred educational programmes which emphasised instrumental support. This interpretation is reinforced by the relative value placed on these groups by men (Q57).

**Survivorship**
**Theme 5: structured follow-up practical information support**
Practical information support was crucial for maintaining the long-term physical and emotional health of caregivers, primarily assisting in preparing families for the future (Q58) and providing opportunity for self-care (Q59). Information on legal matters, entitlements, respite and counselling was instrumental in accessing timely appropriate supports along the entire care continuum (Q60). Hospital social workers were noted as key in the provision of timely information on entitlements, waivers and/or grants and guidance on applications.

Timely provision of practical information by early intervention team members was credited with reduced milestone developmental delay, with most caregivers expressing a preference for in-home visits from these teams in the early years (Q61). Appropriate information on respite care, that outlined the long-term benefits of accessing respite, improved service uptake and engagement (Q62). All caregivers expressed an urgent need for improved access to individual and family counselling (Q63), attributing the lack of psychological support to patient self-harm (Q64) and caregiver hypervigilance (Q65). However, all men expressed profound regret over the lack of engagement with counselling (Q66), with one caregiver emphasising the importance of counselling post-bereavement (Q67). Barriers included masculine stereotypes, the perceived predominant healthcare focus on female mental health, and lack of access to affordable counselling (Q68). Service provision uptake reduced caregiver hypervigilance, reliance on family support, 'over-caring' and social isolation. This is significant given that hypervigilance reduced social networks (Q69) and overall psychological well-being (Q70), which contributed to maladaptive caregiver coping and exhaustion (Q71). In contrast, successful healthcare engagement with families generated positive relationships and encouraged shared decision-making (Q72) which promoted caregiver self-efficacy, self-awareness and acceptance for respite support. Coping appeared most successful when professional support was not stopped prematurely and/or without family discussion, as caregivers associated respite support with maintaining positive relationships with partners and/or unaffected siblings (Q73).

Sharing of practical information between healthcare teams and schools enabled timely applications for additional education staff and physical resources. The sharing of practical information by caregivers within local community groups and/or schools, achieved via conscious networking, served to reduce stigma and increase community understanding of skin disease (Q74 and Q75). This positively benefitted the psychosocial health of themselves, their affected child and the wider community (Q76). Regardless of education level, when practical information support was lacking, caregivers felt alone, frustrated, angry and overwhelmed. Moreover, this reduced caregiver self-care and self-efficacy, and increased the psychosocial impact on both the caregiver and the affected child (Q77).

## Framework for service-provided information needs on the ichthyosis care continuum

This article proposes a theoretical framework (figure 1) to advance an understanding of how service-provided information supports, based on the themes above, can influence self-efficacy and coping ability, in the context of considering the demands and resources of both the caregiver and the caregiving situation. Crucially, this framework considers how the modification of any one element, through feedback loops, can result in a different bidirectional psychosocial impact for the caregiver and the affected child. This circularity demonstrates the important

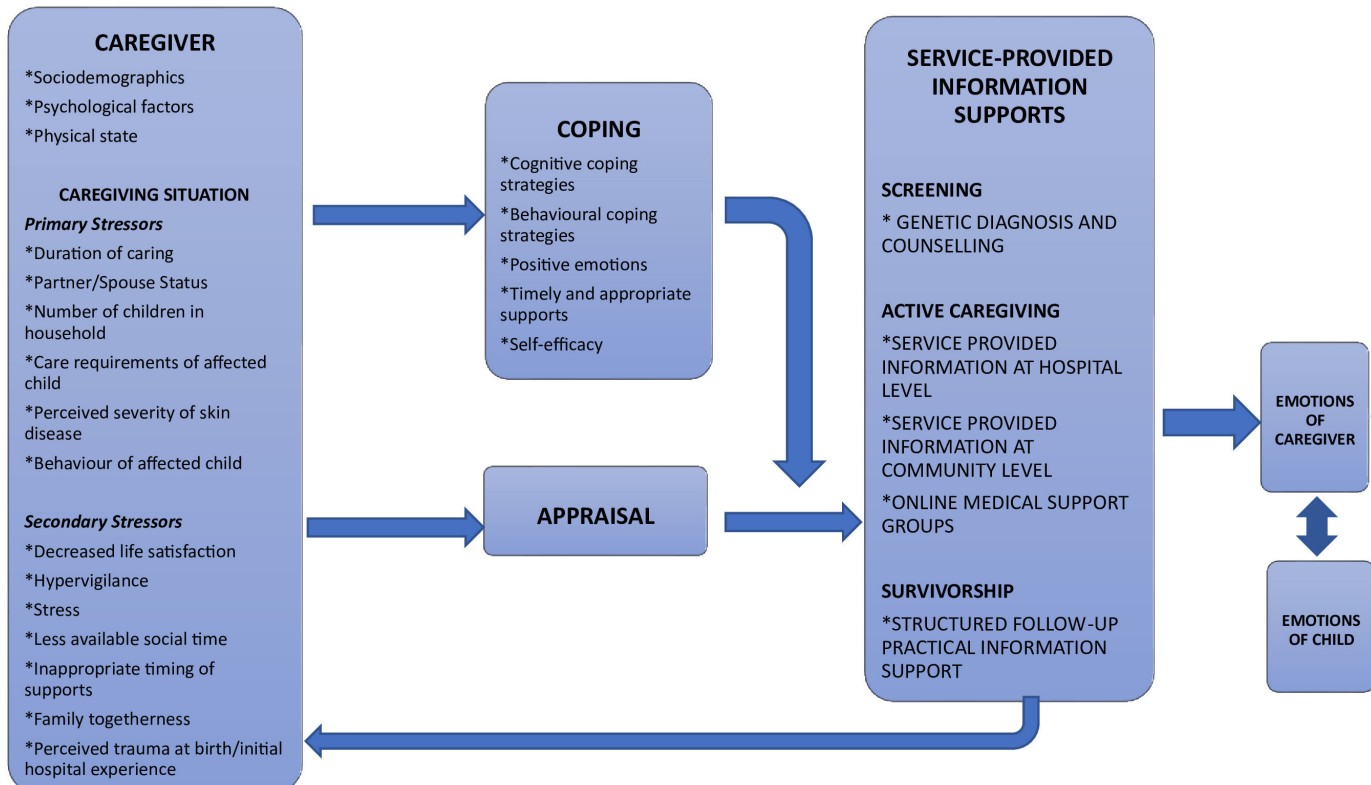

**Figure 1** Theoretical framework for service-provided information needs on the ichthyosis care continuum.

role positive aspects of caregiving can have in counterbalancing negative dimensions, potentially informing strategies for improving the emotional outcomes for everyone in the family.

## DISCUSSION

This study has demonstrated that service-provided information is vital in influencing the self-efficacy, coping ability and psychosocial well-being of dermatological caregivers and their children. Although literature exists on coping for other dermatological conditions,[17–19] this is the first major study to examine the influence of service-provided information support from the caregiver perspective. Our findings provide important insights on existing gaps between caregiver expectations and needs, and how to improve access to and delivery of service-provided information support at key stages of the care continuum. Consistent with Bandura's self-efficacy for caring model,[20] perceived ability to carry out or respond to a range of information-related tasks, including accessing, interpreting and coordinating medical care, appears to be associated with caregivers' ability to adapt positively to their caregiving role, need for disease-specific knowledge and their self-efficacy.[21 22]

Service-provided information support modified caregiver self-efficacy which, via affective and cognitive mechanisms, positively influenced behaviours, appraisal, motivation and emotional robustness of both the caregiver and their child.[23] This is important given that caregivers' self-efficacy has demonstrated a more important role in caregiver distress than the actual demands of caregiving and is associated with a reduced incidence of physical and psychosocial illness.[24] Although research suggests that several dimensions of caregiver self-efficacy have been specifically shaped by the provision of information support,[25] including self-efficacy for coping, this study identified additional dimensions such as self-efficacy for managing hypervigilance, improving social relationships and controlling upsetting thoughts. Our theoretical framework proposes that timely, appropriate information support can counteract the demands of caregiving and promote coping through positive appraisal of resources relating to both the caregiver and caregiving situation. As self-efficacy can change across the care continuum, and in response to specific life experiences, this framework may inform the development and delivery of information supports under five key themes (online supplemental file 5) at three points on the care continuum: screening, active caregiving and survivorship.

Given the prevalence of depression and anxiety in caregivers of children living with chronic skin disease,[26] caregivers lacking in adequate adaptive mechanisms must be supported to protect from and mitigate the deleterious effects of the caregiver distress and depression. In line with recent research,[27] findings confirmed that self-efficacy had little benefit if the environment perpetuated poor health, emphasising the need to address issues such

as service accessibility. Caregivers who perceive they lack appropriate information to respond effectively to caregiving challenges, and those who are more likely to focus on past failures, must be identified and supported. When timely disease and care specific information was provided by hospital and community healthcare teams, care negotiations and treatment adherence were both enhanced. This study reinforces the importance of sharing information through open and honest hospital-community healthcare pathways,[28] particularly if prognostic disclosure is addressed as a once off event as opposed to an ongoing communication.[29] Findings also echoed the importance of receiving consistent information and guidance regarding the safe use of products from healthcare professionals to increase treatment adherence,[30] which was associated with improved caregiver coping ability.[31–33] Similar to findings from a study by Smit *et al*,[34] caregivers coped better when they knew who was responsible for triggering conversations regarding referrals to clinician expertise.

In line with recent research,[27] findings additionally confirmed that self-efficacy specifically shaped by information support had little benefit if the environment perpetuated poor health, emphasising the need to address issues such as service accessibility. Identification of barriers to enhance caregiver 'readiness' for information is particularly crucial for caregivers who are unaware of their need for a timely diagnosis and/or the spectrum of emotional and behavioural reactions that may be experienced in response to genetic testing.[35] Healthcare professionals appear best placed to support caregiver self-efficacy through behavioural, educational and psychosocial interventions that address mastery, verbal persuasion, vicarious experiences and physiological states.[36] The change in preference for obtaining information from dermatology experts during the initial caregiving stage to online ichthyosis support groups may be attributable to the rarity of the disease and the associated lack of healthcare knowledge and/or understanding of ichthyosis and its impact on the family. The way in which caregivers used online support groups is congruent with Wellman's[37] definition of community as 'networks of interpersonal ties that provide sociability, support, information, a sense of belonging and social identity' (p228). Such community identification, as defined by social identify theory[38] is beneficial to caregiver well-being and coping behaviours. Informal caregivers increasingly use online resources for altruism[39] and information gathering and seeking.[40 41] Demonstrating how others cope and adapt to caring for a particular disease can help to validate illness experiences and balance out conflicting offline emotions.[42] Given that caregivers for the more severe subtypes tend to prefer online information support as caregiving duration increases, it seems prudent to signpost caregivers towards online evidence-based information to encourage caregiver self-management and negate potential negative impacts from accessing online medical information.[42 43] Similarly, online support group staff must receive training

to facilitate the development of caregiver self-efficacy. The lack of male online engagement suggests that culture plays an integral role in shaping their perceptions of roles and responsibilities, and could infer that educational programmes which emphasise instrumental support over emotional support may prove a better suited intervention for male caregivers in future. Although improved national healthcare education and training programmes appear crucial for tackling stigma in developing countries, this study additionally highlights the potential role for national media campaigns to tackle the growing social problem, appearance and dissatisfaction in developed countries. In developing countries where the information needs of caregivers are often poorly met, solutions such as tele-dermatology are invaluable for improving self-efficacy.[25]

Subjective evaluation from caregivers highlights the importance of information to counteract potential negative dimensions, which in turn leads to positive emotional and coping outcomes. Caregivers' emotional response to diagnosis was shown to resonate well beyond the diagnostic experience and was seemingly influenced by caregiver experience within the pre-diagnosis stage and the context, delivery and mannerism of how the diagnosis was communicated to the caregiver. Genetic counselling empowered caregivers to access formal supports and effectively plan for the future.[44] Improved positive emotional responses appeared to be associated with improved coping ability, which was influenced by pre-existing perceptions of their own child's risk and disease severity.[45] Regardless of disease subtype or education level, information support positively influenced emotional regulation. This is particularly significant as our findings suggest a bidirectional relationship between the psychosocial well-being of the caregiver and the affected child. Emotional competencies are a promising coping resource for caregivers,[46] supporting the argument for timely, self-report and solution-focused needs e-assessment.[47]

Although there was a female gender bias (4:1), a strength of this study is its international representative caregiver sample which identified five key information themes. Online recruitment from the community may have allowed caregivers to give more candid or less socially desirable accounts of their access to information from formal services. The under-representation of male caregivers and exclusion of caregivers who were not fluent in English during data collection are limitations of this study. The purposive online sampling strategy may negatively influence the transferability of findings to non-users of online peer support groups. Additionally, COVID-19 may have negatively influenced caregiver recruitment and mindset during data collection.

Further research is being conducted by the authors to quantitatively assess the perceived helpfulness of various information supports using both international expert and caregiver groups. Future research should investigate the role of information in caregiver burnout and the mechanisms by which this occurs, with longitudinal studies focusing on the relationship between stressors and coping strategies on the caregiver's emotional experience. Cognitive and behavioural theory identifies many mechanisms that influence responses to health information, which may be investigated in the context of coping ability after receiving relevant information.

In conclusion, caregivers reported improved coping when timely, personalised and appropriate service-provided information support was provided along the care continuum. This study provides concrete evidence of the critical mediating role information support plays in influencing the self-efficacy, coping ability and psychosocial well-being of ichthyosis caregivers. Key recommendations for optimising service-provided information support are included (online supplemental file 6) to give health providers clearer direction on where to focus future efforts to improve information exchange and guide the preparation of support, education and training resources for service providers and healthcare professionals.

**Acknowledgements** The authors would like to sincerely thank each of the caregivers and members of the expert group for their time, contribution and support with this study. A huge thank you to both online support groups (ISG and FIRST) for helping the study with recruitment.

**Collaborators** Not applicable.

**Contributors** GL, MMcL and CW made substantial contribution to the design of this study. CW conducted data collections, coded data, analysed data and drafted the manuscript. GL double-coded all data. GL and MMcL also analysed data. GL, MMcL and CW reviewed drafts, provided valuable inputs and approved the final version. CW acting as guarantor.

**Funding** Bamford Centre for Mental Health and Wellbeing (Ulster University). Award/Grant number is not applicable.

**Competing interests** Abstract was selected for oral presentation (CW) at the 19th Congress of the European Society for Dermatology and Psychiatry (ESDaP) and 2nd Brain Skin Colloqium Conference (London, 2021). Abstract selected for publication by the British Journal of Dermatology. CW was invited to, and attended, the Novartis Global Leadership Forum (Switzerland, 2021) to orally present study findings.

**Patient and public involvement** Patients and/or the public were involved in the design, or conduct, or reporting, or dissemination plans of this research. Refer to the Methods section for further details.

**Patient consent for publication** Consent obtained from parent(s)/guardian(s).

**Ethics approval** Ethical approval was obtained from Ulster University Research Ethics Committee (REC/20/0004), the Ichthyosis Support Group (ISG) and the Foundation for Ichthyosis and Related Skin Types (FIRST). Participating caregivers provided online informed consent. Participants gave informed consent to participate in the study before taking part.

**Provenance and peer review** Not commissioned; externally peer reviewed.

**Data availability statement** All data relevant to the study are included in the article or uploaded as supplementary information.

**ORCID iDs**
Carleen Walsh http://orcid.org/0000-0002-7065-1066
Gerard Leavey http://orcid.org/0000-0001-8411-8919
Marian McLaughlin http://orcid.org/0000-0002-5233-928X

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
