## [Reviewer comments · BMJ Open]

ARTICLE DETAILS

TITLE (PROVISIONAL)	Information provision to caregivers of children with rare dermatological disorders: an international multi-method qualitative study
AUTHORS	Walsh, Carleen; Leavey, Gerard; McLaughlin, Marian

VERSION 1 – REVIEW

REVIEWER	Kentley , Jonathan Royal London Hospital for Integrated Medicine
REVIEW RETURNED	27-Mar-2023

GENERAL COMMENTS	This is an interesting and important study which will help guide provision of information to caregivers of children with a potentially devastating skin condition. The study is well designed, the paper well written and the limitations appropriately acknowledged- I would be happy to accept in the current form.
---

REVIEWER	Parvizi , Mohammad Mahdi Shiraz University of Medical Sciences
REVIEW RETURNED	25-Apr-2023

GENERAL COMMENTS	Dear Editor This is a good and interesting manuscript. There are some recommendations to improve it 1. In the title the authors said rare dermatology diseases, but in the Objective they said ichthyosis. Please clarify this issue. 2. In Table 1, please clarify the role of the parents, it means that the authors interviewed the patient's mother or father and the caregivers' age, too. 3. Please describe the validity and reliability of the used questionnaire. Who confirmed the questions? 4. Please describe the limitations of the study clearer.
---

VERSION 1 – AUTHOR RESPONSE

Reviewer: 1 Dr. Jonathan Kentley , Royal London Hospital for Integrated Medicine	Author(s) Response
This is an interesting and important study which will help guide provision of information	Thank you.

to caregivers of children with a potentially devastating skin condition. The study is well designed, the paper well written and the limitations appropriately acknowledged- I would be happy to accept in the current form.	
Reviewer: 2 Dr. Mohammad Mahdi Parvizi , Shiraz University of Medical Sciences	Author(s) Response
This is a good and interesting manuscript. Below are some recommendations to improve it:	Thank you.
1. In the title the authors said rare dermatology diseases, but in the Objective they said ichthyosis. Please clarify this issue.	Thank you. This qualitative study involved families living with ichthyosis. We too had debated including the term 'ichthyosis' instead of 'rare dermatological disorders' in the title. We based our decision of using the latter in our title due to the rarity of 'ichthyosis' and the inclusion of 'dermatological disorders' as a MeSH term in several electronic databases previously searched in the development of comparable dermatological COS (systematic reviews and meta-analysis). Additionally, the authors and expert group associated with this research felt that the reach/impact of this qualitative study would be greater with a dermatology-specific, as opposed to a disease-specific, term in the title. However, we are very happy to change the title to reflect reviewer suggestion if BMJ Open feel this would be more beneficial.
2. In Table 1, please clarify the role of the parents, it means that the authors interviewed the patient's mother or father and the caregivers' age, too.	Table 1 (Supplemental File 4) has been edited to reflect role of caregiver (mother/father). The expert group, associated with this study, advised us not to include caregivers' age in Table 1 to maintain anonymity, given the rarity of this skin disease and the level of detail already provided in the table. However, to address your comment, we have included a separate table (Table 2) in Supplemental File 4 to provide detail on the caregivers' age. We hope this is satisfactory.
3. Please describe the validity and reliability of the used questionnaire. Who confirmed the questions?	Thank you for highlighting this. Please see details under 'Data Collection' (pg. 6) on how we established trustworthiness. The reference list has been updated to reflect details.

4. Please describe the limitations of the study clearer.	Thank you. Please see page 24 for changes and we hope these have improved the clarity of this section.